# Serotype specific epitopes identified by neutralizing antibodies underpin immunogenic differences in Enterovirus B

Kang Wang [1,2,3,5], Binyang Zheng [1,2,3,5], Li Zhang [3,5], Lunbiao Cui [3,5], Xuan Su[1,3], Qian Zhang [1,3], Zhenxi Guo [4], Yu Guo [1], Wei Zhang [1], Ling Zhu [2✉], Fengcai Zhu [3✉], Zihe Rao[1,2✉] & Xiangxi Wang [1,2✉]

Echovirus 30 (E30), a serotype of *Enterovirus B* (EV-B), recently emerged as a major causative agent of aseptic meningitis worldwide. E30 is particularly devastating in the neonatal population and currently no vaccine or antiviral therapy is available. Here we characterize two highly potent E30-specific monoclonal antibodies, 6C5 and 4B10, which efficiently block binding of the virus to its attachment receptor CD55 and uncoating receptor FcRn. Combinations of 6C5 and 4B10 augment the sum of their individual anti-viral activities. High-resolution structures of E30-6C5-Fab and E30-4B10-Fab define the location and nature of epitopes targeted by the antibodies. 6C5 and 4B10 engage the capsid loci at the north rim of the canyon and in-canyon, respectively. Notably, these regions exhibit antigenic variability across EV-Bs, highlighting challenges in development of broad-spectrum antibodies. Our structures of these neutralizing antibodies of E30 are instructive for development of vaccines and therapeutics against EV-B infections.

[1] State Key Laboratory of Medicinal Chemical Biology, College of Life Sciences and College of Pharmacy and Drug Discovery Center for Infectious Diseases, Nankai University, 300353 Tianjin, China. [2] CAS Key Laboratory of Infection and Immunity, CAS Center for Excellence in Biomacromolecules, Institute of Biophysics, Chinese Academy of Sciences, 100101 Beijing, China. [3] Jiangsu Provincial Center for Disease Control and Prevention, Jiangsu Road, 210009 Nanjing, Jiangsu, China. [4] Cryo-EM core facility, School of Life Science, Peking University, 100871 Beijing, China. [5] These authors contributed equally: Kang Wang, Binyang Zheng, Li Zhang, Lunbiao Cui. ✉email: lingzhu@ibp.ac.cn; jszfc@vip.sina.com; raozh@xtal.tsinghua.edu.cn; xiangxi@ibp.ac.cn

The *Enterovirus* genus, one of the most populous in the family *Picornaviridae*, consists of four human enterovirus species (EV-A, B, C, and D), five animal enterovirus species and three human rhinovirus species afflicting millions of people worldwide annually[1]. Echovirus 30 (E30), one serotype of the species EV-B, has emerged as one of the leading etiological agents of aseptic meningitis and caused extensive seasonal and periodical outbreaks throughout Europe, Asia and South America in recent years[2–4]. Furthermore, E30 is reportedly associated with other diseases like viral encephalitis, acute flaccid paralysis, hepatitis, and even acute diarrhea[5]. Recently, E30 infections have also been shown to cause hand-foot-and-mouth diseases (HFMD) in children, possibly due to genomic recombination events with EV-As[6,7]. The prospect of genetic recombination and antigenic drift together with the recent widespread circulation, point to the potential risk of E30 evolving into a virulence-enhanced pathogen that could endanger the global human health[8]. Currently, there are no approved vaccines or antiviral therapies available for treating infections caused by EV-Bs.

The host humoral immune response acts as a major defense against viral infections[9]. Vaccines prime these defenses by eliciting protective neutralizing antibodies (NAbs)[10]. Passive immunization has also been demonstrated to be effective in curing diseases[11–14]. A deep understanding of the molecular basis for viral neutralization by antibodies and the identification of key viral epitopes would aid in the development of rationally designed vaccine and antiviral drugs. Such strategy, however, has not yet been explored for EV-Bs. It is well known that many picornaviruses utilize two types of receptors, attachment and uncoating, to initiate efficient cellular entry processes, including viral attachment, endocytosis, internalization, uncoating and genome release [see coordinated submission by Wang et al.[15]][16,17]. These processes are generally accompanied by a cascade of structural rearrangements of viral capsid proteins initiated and mediated by receptor binding as well as specific microenvironments [see coordinated submission by Wang et al.[15]][6,18–26]. Neutralizing antibodies could possibly target different steps of viral infection, including blocking viral attachment to the cellular receptor, interfering with viral uncoating via over-stabilizing or destabilizing the virus and physically damaging and/or aggregating the virus[12,27,28]. Due to large gaps in our knowledge concerning the immunogenic features and key epitopes of E30, questions about whether its NAbs can cross-react with other EV-Bs or cross-protect against infections by other EV-Bs remain unanswered.

Here we elicited two highly potent serotype-specific NAbs, referred to as 6C5 and 4B10, both of which could neutralize E30 infection efficiently by blocking viral binding to its two types of receptors. Atomic structures of E30 in complex with 6C5/4B10 reveal the nature of the binding modes and locations of epitopes targeted by these two antibodies. The key epitopes distinguishing E30 from other EV-Bs, partially overlap with the footprints of the receptors on the viral capsid and highlight the area of antigenic variability in EV-Bs. Information about the area of antigenic variability is an important consideration during rationally designing effective multivalent vaccines and broad-spectrum antiviral therapeutics.

## Results

### Characterization of anti-E30 NAbs 6C5 and 4B10. Two antibodies, 6C5 and 4B10, were generated by immunizing BALB/c mice with formaldehyde-inactivated E30 mature virions. To investigate the serotype specificity of 6C5 and 4B10, we propagated, purified Echovirus 3 (E3), Echovirus 6 (E6), Echovirus 11 (E11) and Coxsackievirus B3 (CVB3) virions, and separately examined their binding abilities to each antibody by enzyme-

linked immunosorbent assay (ELISA). The ELISA experiments showed that both 6C5 and 4B10 bind E30, but neither react with E3, E6, E11 or CVB3, suggesting that both antibodies are serotype specific (Fig. 1a). Surface plasmon resonance (SPR) assays demonstrated that 6C5 and 4B10 both exhibit tight binding to E30 with affinities of 1.51 and 2.88 nM, respectively (Fig. 1b). To explore whether these two antibodies recognize different or the same patch of epitopes, we performed a competitive SPR assay (see "Methods") and the result indicated that the binding of one antibody blocks the attachment of the other (Fig. 1c), which raises the possibility of 6C5 and 4B10 binding to the same epitope or at least partially overlapped epitopes. In line with the binding results, cell-based viral neutralization investigations revealed that both antibodies could efficiently neutralize E30 infection as intact antibodies or Fab fragments with 50% neutralizing activities in the nanomolar ranges, but neither 6C5 nor 4B10 could protect against other EV-Bs (Fig. 1d, e). To further verify the potency of the two antibodies, groups of murine E30 antisera with exceptionally high titers were subjected to a competitive binding efficiency test against 6C5 and 4B10. These sera showed high blocking rates against 6C5 and 4B10 binding to the E30 virions ranging from 60% to 80%, reflecting that epitopes of 6C5 and 4B10 are dominant in antisera (Fig. 1f, g). In general, functional receptor attachment is capable of dissociating viral capsid protein interactions to trigger genome release[3,16,17,29,30]. However, both intact antibodies and Fab fragments of 6C5 and 4B10 destabilized slightly E30 virions by 1–3 °C (Fig. 1h and Supplementary Fig. 1). Given that E30 virions continues to exist as mature virion at physiological temperatures even after slight destabilization by 6C5 and 4B10 (Supplementary Fig. 2), destabilization of the virus is unlikely to be their mechanism of neutralization.

**Both 6C5 and 4B10 block viral binding to its receptors**. As with many other EV-Bs, E30 is believed to utilize two types of receptors to gain entry into host cells: CD55 for viral attachment; and FcRn for viral uncoating[16,31,32]. To verify if these two molecules act as receptors and directly bind to E30, the extracellular domains of human CD55 and FcRn were prepared for the binding assay. SPR assays indicated that both CD55 and FcRn interact with the E30 virion with similar binding affinities of ~2 μM, about 1000-fold lower than those of 6C5 and 4B10 with E30 (Fig. 2a). To investigate whether 6C5 or 4B10 interferes with the binding of E30 to CD55/FcRn, we performed four sets of competitive binding assays by exposing the E30 virions to the receptor first and then to the antibodies or the other way around. E6 NAb 1A1 and EVD68 receptor ICAM5[33] molecules were used as negative controls and neither showed any binding to E30 (Fig. 2b, c). In contrast, binding of 6C5 or 4B10 completely blocked the attachment of the two types of receptors to E30. Moreover, both of the receptors could be displaced from E30 and replaced by either 6C5 or 4B10 (Fig. 2b, c). To further verify these results in a cell-based viral infection model, real time reverse transcription polymerase chain reaction (RT-PCR) was carried out to quantify the amount of virus remaining on the host cell surface after exposure to antibodies either before or after viral attachment to cells at 4 °C. Consistent with the competitive binding results, both 6C5 and 4B10 efficiently prevented E30 attachment to the cell surface and could displace the viral particles that had already attached to the cell surface in a dose-dependent manner (Fig. 2d). Intriguingly, when used together, the anti-viral effect of 6C5 and 4B10 as a result of the disruption of the virus-receptor interaction, was greater than those of the sum of their individual anti-viral activities (Fig. 2d). Thus, the NAbs augment each other's anti-viral activity, functioning in a complementary manner.

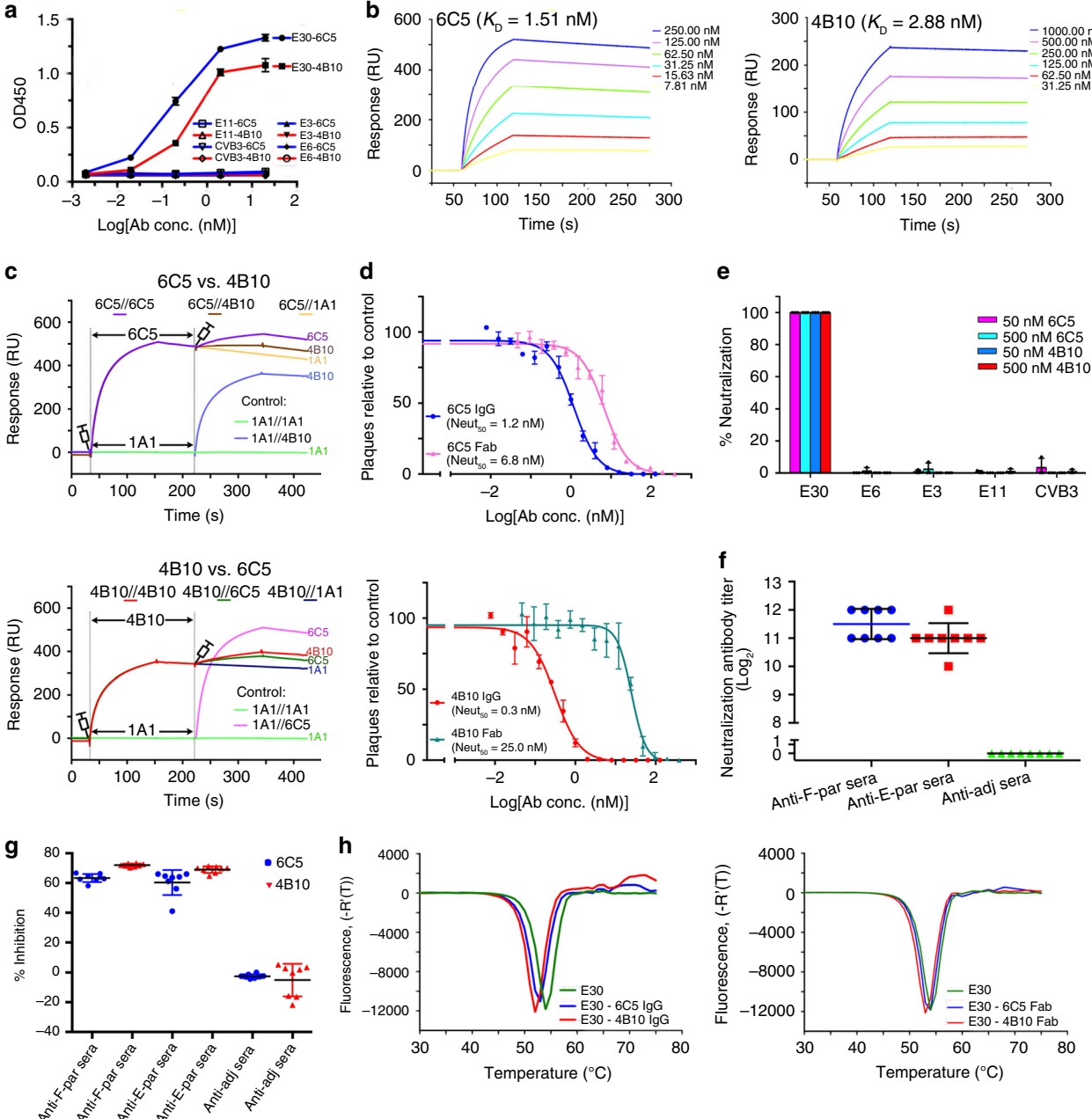

**Fig. 1 Characterizations of the MAbs 6C5 and 4B10. a** Dose-dependent binding analysis of MAbs (6C5 and 4B10) against representative enterovirus B members by ELISA. Each plot represents the mean of OD450 values from triplicate wells. Error bars represent mean ± SD. **b** BIAcore SPR kinetic profile of MAb 6C5 (left) and 4B10 (right) against E30 virus. The binding affinity $K_D$ (equilibrium dissociation constant, $K_D = K_d/K_a$, where $K_d$ and $K_a$ represents the dissociation rate constant and association rate constant, respectively) values were obtained by using a series of MAb concentrations and fitted in a global mode in each sensorgram. **c** Competitive binding between 6C5 and 4B10. In the upper panel, MAb 6C5 was injected first, followed by the second injection of 4B10 or 1A1 specific to E6 as control. In the lower panel, 4B10 was injected first, which was followed by 6C5 or 1A1; the control groups are carried out as the other lines exhibit. **d** Neutralization of E30 by 6C5 (top) and 4B10 (bottom) using plaque-reduction neutralization test (PRNT). Neut50 values indicate concentration of antibody required to neutralize fifty percent of the viral titer. The Neut50 of 6C5 IgG, 6C5 Fab, 4B10 IgG and 4B10 Fab were 1.2 nM, 6.8, 0.3 and 25.0 nM, respectively. **e** Neutralization test of MAbs against representative enterovirus B members by PRNT. An amount of 50 nM/500 nM of 6C5 or 4B10 was used to test whether these two antibodies could cross-neutralize other enterovirus B members. **f** Titration of sera from mice immunized with E30 F-particle or E-particle. Both antisera could highly neutralize E30, while the sera from adjuvant-immunized mice failed to protect RD cells from E30 infection. **g** Competitive ELISA. The antisera against E30 F-particle, or E-particle or adjuvant (as control) was first added to E30 to try to block the binding of E30 with 6C5 or 4B10 added later. **h** Stabilities of E30 upon addition of 6C5 or 4B10. The whole antibodies of 6C5 and 4B10 in complex with E30 (left), as well as their corresponding Fab parts in complex with E30 (right) were mixed with SYTO9 to detect the exposed viral RNA when a heat gradient was applied. All results in **a**, **d**–**g** were expressed as mean ± SD.

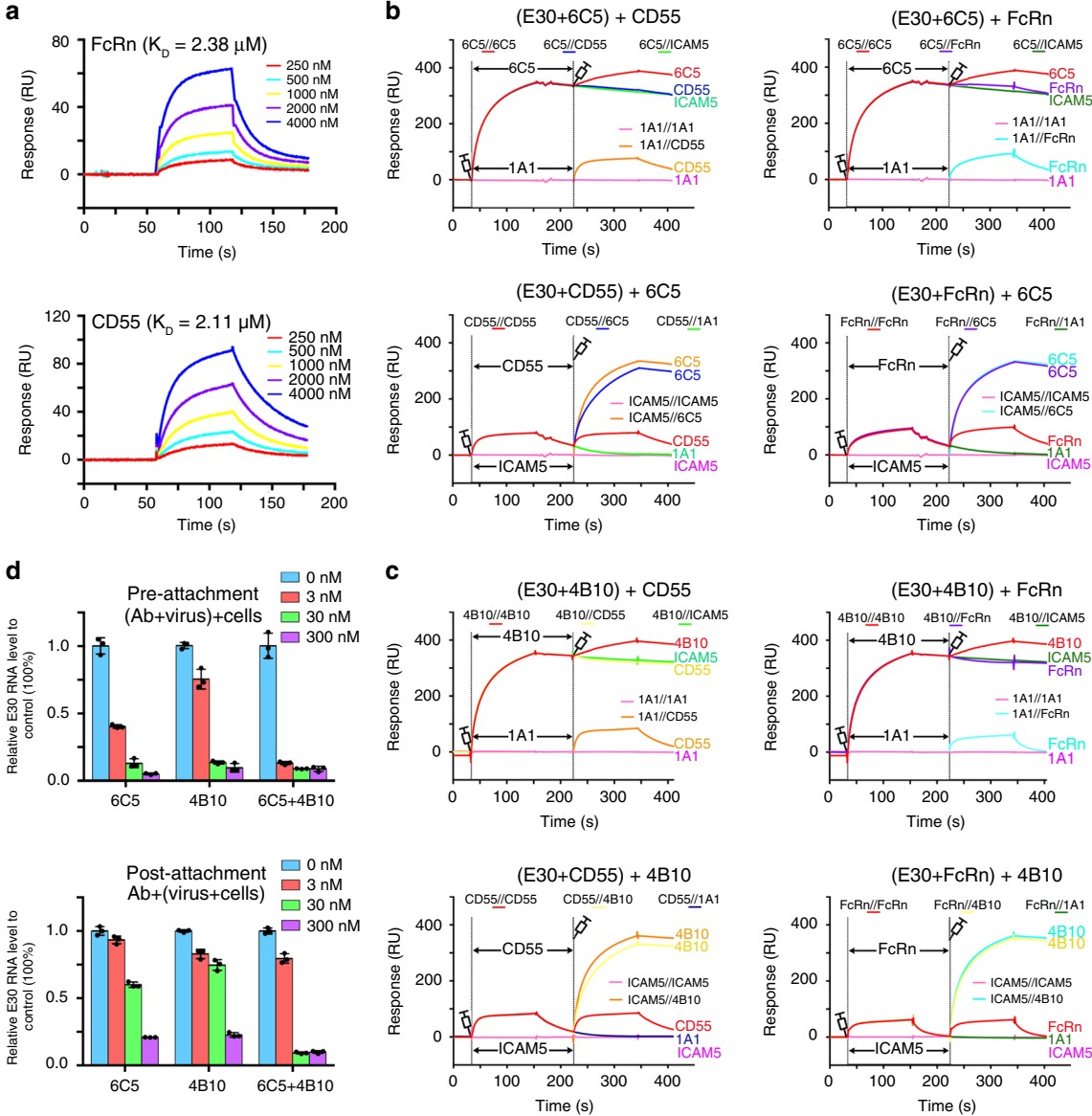

**Fig. 2 Blocking of viral attachment receptor CD55 and uncoating receptor FcRn by 6C5 and 4B10. a** BIAcore SPR kinetics of FcRn or CD55 binding to E30. The RU curves are fitted globally to calculate the $K_D$ value for FcRn and CD55. **b**, **c** Competitive binding between 6C5 or 4B10 and CD55 (left) or FcRn (right). In the upper two diagrams of each panel, 6C5/4B10 was injected first, followed by either CD55, or FcRn, or ICAM5, the uncoating receptor for EVD68; in the lower two diagrams of the counterparts, CD55 or FcRn was injected first, followed by either 6C5, or 4B10, or 1A1. The related control groups were carried out as the other curves show. **d** Amount of virions remaining on the cell surface, as detected by real-time PCR, when exposed to 6C5 (left) or 4B10 (middle) or the mixture of 6C5 and 4B10 (right) before or after the virions attach to RD cells and the results here were expressed as mean ± SD.

**Structures of E30 in complex with its NAbs 6C5 and 4B10**. To define the key epitopes and atomic determinants of the interactions between E30 and its two NAbs precisely, structural investigations of E30 in complex with Fab fragments from 6C5 and 4B10 were conducted. Cryo-EM micrographs of the formaldehyde-inactivated E30 virions (see the coordinated submission by Wang et al.[15]) in complex with 6C5/4B10 Fab fragments were recorded using an FEI Titan Krios electron microscope equipped with a Gatan K2 Summit detector (Supplementary Fig. 3). The cryo-EM structures of E30-6C5 and E30-4B10 complexes were determined at resolutions of 3.1 and 3.7 Å, respectively (Fig. 3a and Supplementary Fig. 3). The cryo-EM electron density maps were of high quality; allowing us to build the models of these two complexes (Fig. 3b). E30 capsid proteins exhibit no notable conformational changes upon binding to 6C5 or 4B10 with RMSDs of 0.3 and 0.4

Å, respectively, between the 6C5/4B10 Fab bound and unbound states of E30.

The 6C5 Fab fragment binds to the E30 viral surface within the pentameric building blocks at a site near the fivefold axis (Fig. 3c). The position of this binding site is similar to those observed previously for 11G1 antibody bound to Enterovirus D68 (EVD68) and NAb 1D5 bound to Coxsackeivirus A6 (CVA6)[34]. Five 6C5 Fab molecules attach to the north rim of the canyon that constructs the "star-shaped" protrusions surrounding the mesa [see coordinated submission by Wang et al.[15]]. When compared to the 6C5 Fabs, the Fabs of 4B10, contacting the south wall of the canyon, are comparatively more spread out (Fig. 3d). Interestingly, the 6C5 and 4B10 Fabs adopt distinctly different configurations upon binding to the E30 (Fig. 3c–e). When viewed down the fivefold axis, individual 4B10 Fabs stand

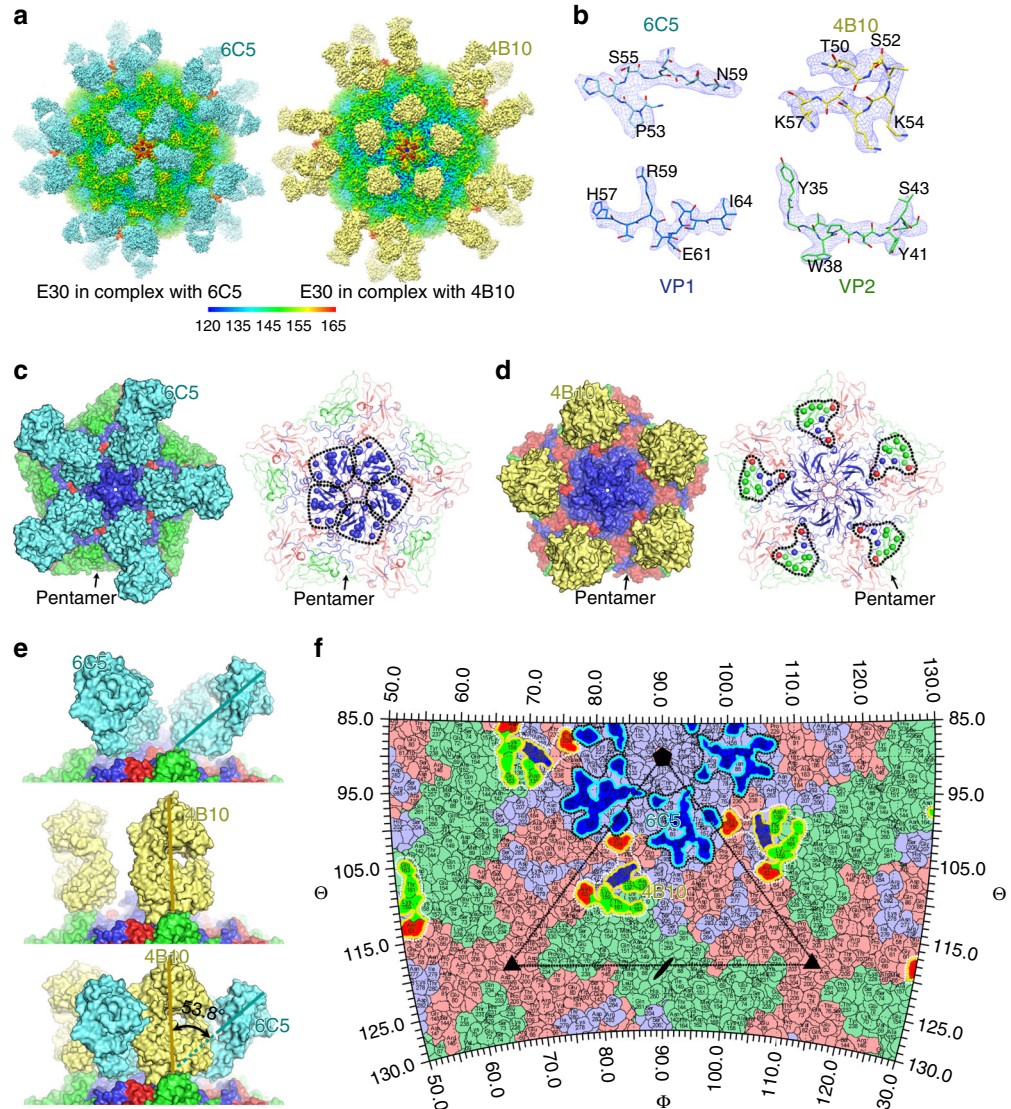

**Fig. 3 Cryo-EM structures of E30 in complex with 6C5 Fab or 4B10 Fab. a** Surface representations of E30-6C5-complex (left) and E30-4B10-complex (right). The viral parts of both complexes are rainbow-colored as the color bar below shows; the 6C5 Fab and 4B10 Fab are colored in cyan and yellow, respectively. **b** Electron density maps for representative sections of VP1 and 6C5 (left) from E30-6C5-complex, and sections of VP2 and 4B10 (right) from E30-4B10-complex. **c, d** Fab occupancy (left) and epitopes (right) of 6C5 (c) and 4B10 (d) on a viral pentamer. The pentamers are shown as surface (left) or cartoon (right) in the signature colors (VP1, blue; VP2, green; VP3, red), while Fabs are colored in the same scheme as in 3a. The epitopes of 6C5 (left in 3c) and 4B10 (right in 3d) are shown as spheres and those from one protomeric unit are circled by black dotted lines. **e** Side views of two Fabs bound to a pentamer. The same color scheme is applied as above. When the E30-6C5 (top) and E30-4B10 (middle) are superposed, the complex (bottom) generated shows an angle of 53.8° between 6C5 and 4B10. **f** Footprints of 6C5 and 4B10 on the sterographic projection of E30. Residues of VP1, VP2 and VP3 are colored in pale blue, pale green and pale red, respectively; residues involved in binding with 6C5 and 4B10 are shown in brighter colors corresponding to the protein subunit they belong to. The footprints of 6C5 and 4B10 on each protomeric unit are outlined in cyan and yellow, respectively, and circled in black and white dotted lines, respectively. One icosahedral asymmetric unit with five-, three- and twofold icosahedral symmetry axes are marked out.

vertically in an in-canyon attachment mode, resembling the strategy used by the uncoating receptor FcRn for its association with E30 [see coordinated submission by Wang et al.[15]]. In contrast, the 6C5 Fab inclines backward by ~50° crossing the canyon from close to a fivefold axis toward an adjacent threefold axis (Fig. 3c–f), covering a vast region of the surface, despite having a similar interaction area of ~800 Å$^2$.

In the 6C5 Fab and 4B10 Fab bound E30 structures, variable domains of the light chain and heavy chain contribute ~39%, 61%, and ~92%, 8% of the protein-protein interactions, respectively, with 6C5 predominantly contacting VP1, whereas 4B10 largely binding to VP2 (Fig. 4a). The interaction patch in 6C5 comprises all six complementary determining regions

(CDRs): L1 (residue 30 and 31), L2 (residue 51), L3 (residue 90), H1 (residue 32 and 33), H2 (residue 54 and 55), H3 (residues 99-101) and the light chain framework region (LFR, residue 33 and 48). The epitope recognized by 6C5 contains 11 residues, primarily locating in the VP1 BC loop (E82, K83, V84, D86, E87, D89, Y91), VP1 DE loop (T130), VP1 EF loop (K156, E159), and the VP1 HI loop (T229) (Fig. 4b). However, the epitope of 4B10 mainly includes residues 137, 138, 159, 161 and 163 of VP2 EF loop, residues 260 and 268 of VP1 C-terminal loop and residue 234 of VP3 C-terminus (Fig. 4b). Tight bindings between the antibodies and E30 are chiefly due to extensive hydrophilic interactions, including hydrogen bonds and salt bridges (Supplementary Table 2 and Table 3). Although the footprints of 6C5

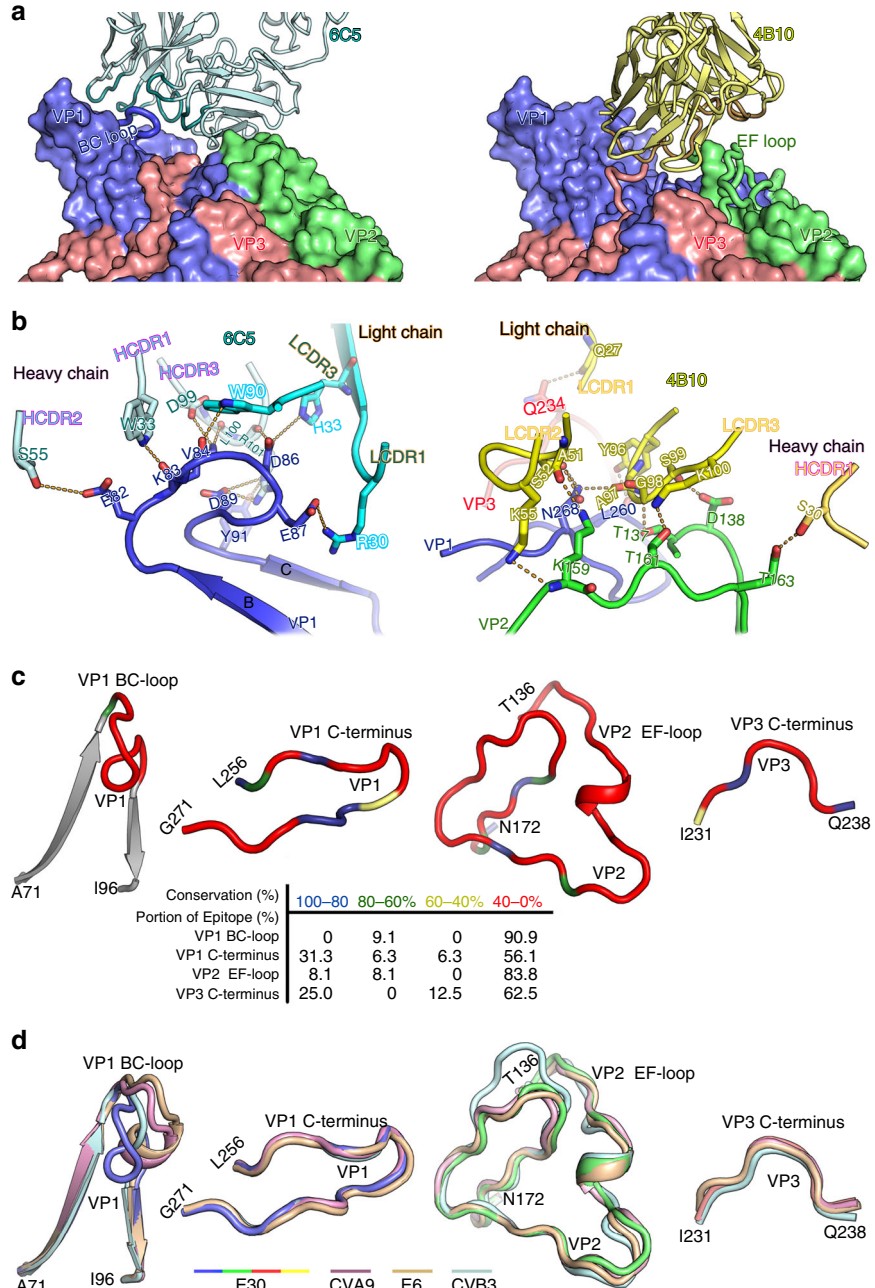

**Fig. 4 Interactions between E30 and 6C5/4B10. a** Platform for Fab (6C5, left and 4B10, right) binding with E30. Loops from E30 involved in interactions and the Fab parts are shown as cartoon, while the remaining parts of E30 are shown as surface. Also, the parts from 6C5 and 4B10 interacting with E30 are colored in dark cyan and dark yellow, respectively. The color scheme is as in Fig. 3c. **b** Binding interface between E30 and 6C5 (left), as well as E30 and 4B10 (right). Residues involved in the binding are shown as sticks and hydrogen bonds are shown as orange dashed lines. **c** Sequence conservation analysis. The loops from E30 that are involved in interactions between E30 and 6C5 (VP1 BC loop), as well as E30 and 4B10 (VP1 C-terminus, VP2 EF loop and VP3 C-terminus) are aligned with those of other enterovirus B members (E1, E6, E7, E11, E18, CVB3, CVA9), and colored according to sequence conservation as listed in the table below. **d** Structure conservation analysis. The same loops in 4c colored in the signature color scheme are superposed with their counterparts from E6 (colored in wheat), CVB3 (colored in pale cyan), and CVA9 (colored in pale magenta).

and 4B10 on the E30 surface do not overlap completely, these two Fab fragments clash sterically due to proximity and the inclined posture of 6C5. These structural observations are consistent with the results of the competitive binding studies (Fig. 1c).

**Serotype specific binding modes for 6C5 and 4B10.** Functional characterization of NAbs 6C5 and 4B10 revealed that these antibodies are serotype specific, suggesting antigenic differences

in the different subgroups of EV-Bs. A number of studies have revealed a role for several exposed loops engaged in the construction of the canyon of *Enteroviruses* in the serotype-specific differences [see coordinated submission by Wang et al.[15]][7,35]. These loops include VP1 BC, GH loops, VP2 EF loop and VP1 C-terminal loop, which are also often mapped as neutralizing epitopes[11,27,28,36]. As a major structural marker, the VP1 BC loop not only contributes significantly to distinguishing E30 from other EV-Bs, but it is also the most divergent region with regards

to primary sequence within EV-Bs (Fig. 4c and Supplementary Fig. 4). Overall, excluding 10% of the binding area contributed by conserved residues, the average conservation is only 26% in the 6C5 epitope (Fig. 4c and Supplementary Fig. 4). The specificity of VP1 BC loop both in sequence and configuration and it's recognition by 6C5 explain the serotype-specificity of 6C5 (Fig. 4c, d). In contrast to 6C5, 4B10 buries 495 Å$^2$ of the VP2 surface by interaction with VP2 EF loop and 200 Å$^2$ of VP1 as well as 110 Å$^2$ of VP3 via association with their C-terminal loops. Unlike the VP1 BC loop, the backbone Ca atoms of VP2 EF loop, VP1 C-terminal loop and VP3 C-terminus of E30 adopt similar conformations as those observed in other EV-Bs. However, the primary sequence of these regions varies across EV-Bs, indicating that the side-chain dependent interactions play critical roles in the recognition of the E30 antigenic determinants by 4B10 (Fig. 4c, d). Unexpectedly, the VP1 GH loop, harboring the widely reported major antigenic sites in EV-As[27,37,38], is unlikely to contribute to the key epitopes in E30 due to the failure in obtaining NAbs targeting this loop despite many trials. In general, protective antibody diversity, such as 6C5 and 4B10 elicited by E30, is an important feature of the adaptive immune system, wherein the system protects hosts against viral infection by producing diverse protective antibodies. Since E30 elicits production of strong neutralizing antibodies like 6C5 and 4B10, it qualifies as a reasonable vaccine candidate [see coordinated submission by Wang et al.[15]].

**Structural superimposition studies reveal steric clashes between 6C5/4B10 and receptors.** Competitive binding assays demonstrated the abilities of 6C5 and 4B10 to effectively abrogate the interactions between E30 and its receptors FcRn and CD55 (Fig. 2b–d). Atomic structures of E30 in complex with FcRn/CD55 reveal that FcRn inserts into the viral canyon depression through primarily binding to VP1 GH, VP2 EF and parts of VP1 BC loop, while CD55 lies outside the canyon, adjacent to the "south wall" of the viral canyon [see coordinated submission by Wang et al.[15]]. FcRn presents a classical "in-canyon" recognition mode for most uncoating receptors, while CD55 exhibits a representative attachment strategy for many attachment receptors in picornaviruses. Superpositions of the E30-FcRn/E30-CD55 and E30-6C5 Fab/E30-4B10 Fab complex structures showed clashes between the two receptors and 6C5/4B10. Notably, the superimposition analysis reveals that 4B10 targets the canyon in a manner similar to FcRn (Fig. 5a, b). A number of receptors have been shown to insert themselves inside the viral canyon, whose conserved residues can, therefore, slip under the radar of the immune system, like KREMEN1, FcRn, and CD155 (major receptors for EV-As, EV-Bs, and EV-Cs, respectively)[16,31,39–41]. Unexpectedly, these receptor binding residues are remarkably non-conserved across receptor-dependent viruses[42,43]. Of the FcRn-binding residues, only VP1 Gly151, Gly207, and VP3 Gln238 are conserved and involved in tight interactions with FcRn. Most essential conserved residues for receptor binding present weak side-chain recognition signals, but control the local protein configuration, indicating that receptor binding is largely driven by side-chain independent interactions [see coordinated submission by Wang et al.[15]]. Such a strategy is likely to mitigate the constraints imposed by antigenic variation in receptor binding. For a relatively blunt antibody, the binding sites are usually outside the canyon – the "uncoating receptor" footprint, widely scattered in the most exposed regions with serotype-specific configurations, as those observed for E30 6C5, EV71 D6 and CVA6 1D5 (Figs. 5a, b and 6)[27,34]. In these cases, the neutralizing epitopes, at the most, partially overlap the receptor binding sites, and thus these viruses may evolve by mutating residues in the

non-overlapped regions, thereby abrogating neutralization by their antibodies without imping on receptor recognition. However, 4B10 directly inserts into the canyon and reaches the broad back region (Fig. 5c). Footprints of FcRn and 4B10 on the E30 surface reveal an overlapped patch with an area of ~100 Å$^2$. Thus, the ability of the NAbs in preventing E30 from binding receptors can be attributed to steric clashes arising out of proximity and partially overlapping binding sites.

**Discussion**

Strongly neutralizing antibodies often recognize tertiary/quaternary structure-dependent epitopes which are formed as a result of unique arrangements of capsid proteins (VP1-VP3) constructing specific structural features on the viral surface. In enveloped viruses, flaviviruses, for example, cross-reactive antibodies primarily targeting determinants around the fusion loop of the envelope protein have been widely generated[11,44–47]. However, antibodies capable of neutralizing multiple serotypes and species of enteroviruses (or picornaviruses) have been rarely reported; several exceptions being the limited cross-neutralization activity of MAbs against 2-3 poliovirus (PV) types via binding to the canyon[36,48] and human rhinovirus serotypes by targeting buried VP4[49], respectively. The reasons for the inability to elicit broad-spectrum neutralizing antibodies against EVs can be inferred from the relatively less conservation of the structural features and the varied mechanisms of entry between the different viral subtypes. Subtypes of EVs are known to use different receptors, distinct uncoating patterns, and exhibit diverse antigenic features. Interestingly, however, almost every member of the EV genus possesses the canyon where various types of uncoating receptors often attach; albeit with slightly different morphologies. Perhaps correlated with this, the available cross-neutralization antibodies have been shown to bind PV in the vicinity of the canyon[48]. These previously reported observations together with our current structural analysis suggest antibodies targeting epitopes "inside the canyon" have the potential to confer cross-protection against EVs and these "in-canyon" antibodies may largely mimic receptor binding mode. In this study, we have identified two highly efficient E30-specific NAbs (6C5 and 4B10). Structural and immunological investigations reveal that 6C5 and 4B10 target the north rim of the canyon and in-canyon regions, respectively, expanding our understandings of the epitopes for EV-Bs.

Previous cryo-EM investigations of virion-NAbs complexes in picornaviruses have identified a number of neutralizing epitopes, which are scattered around four major patches: (1) epitopes locating at the north rim of the canyon around the fivefold axis that are recognized by E30 6C5, CVA6 1D5, and EVD68 11G1[14,34]; (2) epitopes positioned at the south rim of the canyon surrounding the twofold axis targeted by EV71 D6, EV71 22A12, EV71 D5, and CVA10 2G8;[27,38,50] (3) epitopes near the threefold axis, for example, those targeted by EV71 E18, EV71 A9, EVD68 15C5 and HRV-B14 C5;[14,20,27,28] (4) epitopes inside the canyon, defined by E30 4B10 and PV A12[36] (Fig. 6). Patches 1 and 2, partially overlap or are adjacent to the binding sites of cellular receptors, but generally do not include necessarily conserved residues for receptor binding. Therefore, although the NAbs elicited by these two patches can sterically clash with receptors and prevent the attachment of the virus to the receptor, the virus can escape host immune responses through self-induced antigenic variations that do not perturb the receptor recognition. Interestingly, NAbs bound to patch 3 are capable of destabilizing viral particles, interfering with uncoating (e.g. triggering the conversion from mature virions to uncoating intermediate particles to release viral genome) or even causing physical damage to the

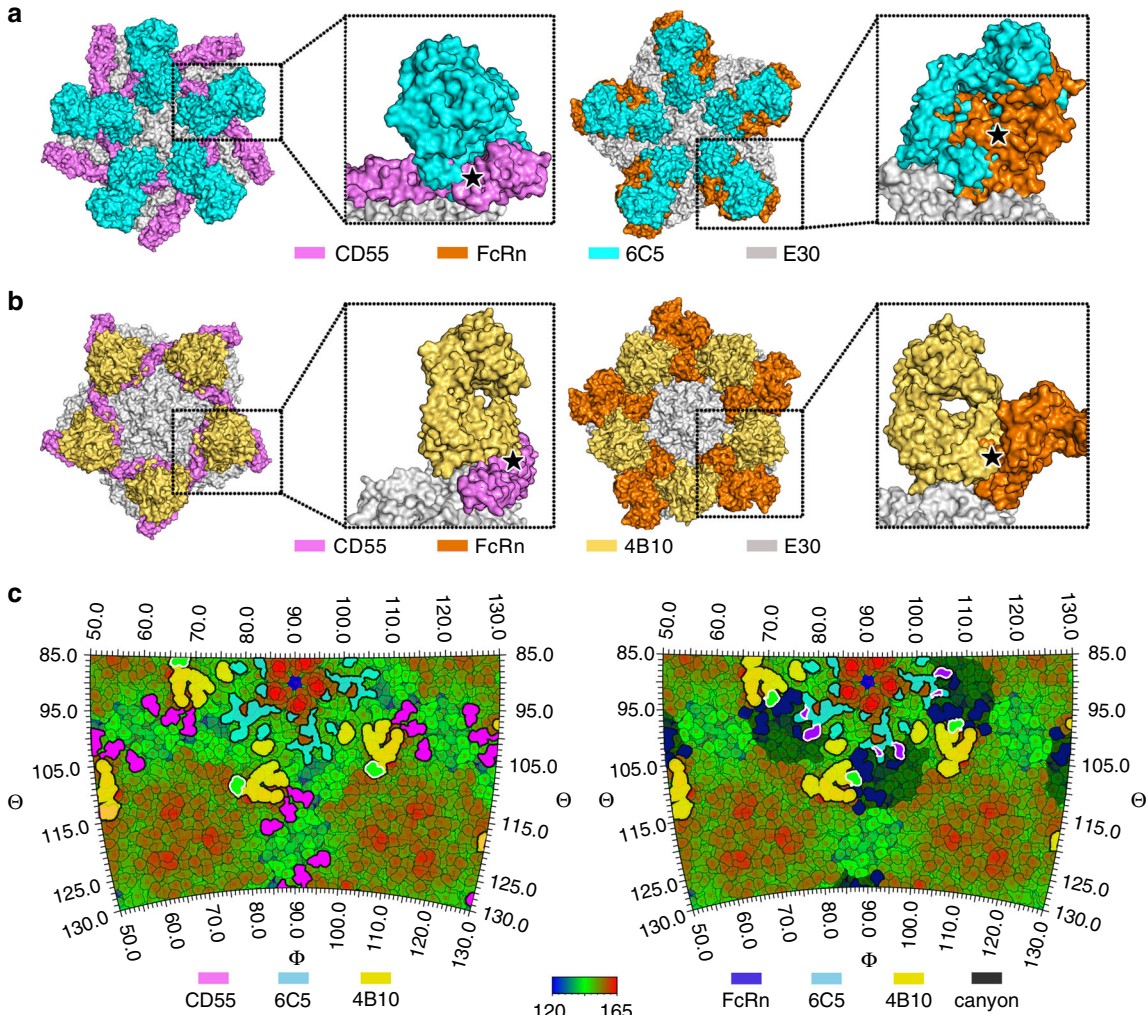

**Fig. 5 Mechanism of neutralization of E30 by NAbs 6C5 and 4B10. a, b** Clashes between Fab 6C5 (a)/4B10 (b) and E30 cellular receptors – CD55 (left)/ FcRn (right). The viral pentamer, Fabs 6C5 and 4B10, and receptors CD55 and FcRn are colored in grey, cyan, yellow, magenta, and orange, respectively. When E30-receptor-complex is superposed with E30-Fab-complex, clashes between each Fab and each receptor are prominent and marked with star symbols. **c** Roadmap exhibiting the footprints of FcRn, CD55, 6C5 and 4B10 on the viral surface. The footprints of CD55, 6C5 and 4B10 are colored in magenta, cyan, yellow, respectively in the left map, and the footprints of FcRn, 6C5 and 4B10 are colored in blue, cyan, yellow, respectively, in the right map, where canyons are shaded by light shadows. Overlapped footprints between receptors and 4B10 are colored in green and outlined with white lines, while the ones between receptors and 6C5 are colored in purple and outlined with white lines.

virions. Patch 4 expands our original understanding of the nature of the antibody-epitope interactions in EVs by unveiling an unknown fact - a relatively blunt antibody can insert itself into the canyon and possibly seize the essentially conserved residues for receptor recognition—hitting the bull's-eye for abrogating virus-receptor interactions in the process. Receptor usage, closely correlated with the canyon morphology, determines cell tropism, and drives viral classification. Theoretically, epitopes in the vicinity of or inside the canyon have the potential to elicit antibodies that cross-neutralize a specific receptor-dependent group of viruses. Despite the serotype specificity of 4B10, receptor mimic side-chain independent interactions targeting the inner surface of the canyon, in particular the necessarily conserved residues VP1 Gly151, Gly 207, and VP3Gln 238 in E30, could be rationally designed and generated to obtain broad-spectrum NAbs against EV-Bs. The molecular basis of the 6C5 and 4B10 epitopes revealed in this study not only provides interesting targets for structure-based multivalent vaccine design, but also highlights the possibility of antibody-based therapeutic inventions.

## Methods

**Ethics statement.** Animal care and studies were carried out in strict accordance with the guidelines and regulations of the Guide for the Care and Use of Laboratory Animals of the Ministry of Science and Technology of China. All protocols and procedures were approved by the Animal Welfare and Ethics Committee at Jiangsu Provincial Center of Disease Control and Prevention.

**Particle purification.** For the purification of E30 particles. RD cells were inoculated with the E30 strain at an MOI = 0.1 and incubated at 37 °C for 18–24 h. When 95% of the cells developed prominent cytopathic effect (CPE), the culture was harvested and subjected to a low speed centrifugation (at $1500 \times g$ for 30 mins) to remove the cell debris, and then the supernatant was followed by high speed centrifugation (at $120,000 \times g$ for 2 h) to obtain the pellet, which was further applied to a continuous 15–45% (w/v) sucrose density gradient centrifugation (at $104,100 \times g$ for 3.5 h) for harvesting the virus. Finally, the fractions containing full particles and empty particles were pooled and dialyzed against buffer phosphate-buffered saline (PBS) (ThermoFisher, catalog #10010023) (see coordinated submission by Wang et al.[15])[51–53].

**Negative staining.** The intact E30 F particles, heat-treated E30 F particles, E30 F particle in complex with 6C5 Fab or 4B10 Fab with a ratio of ~1:200 and dropped

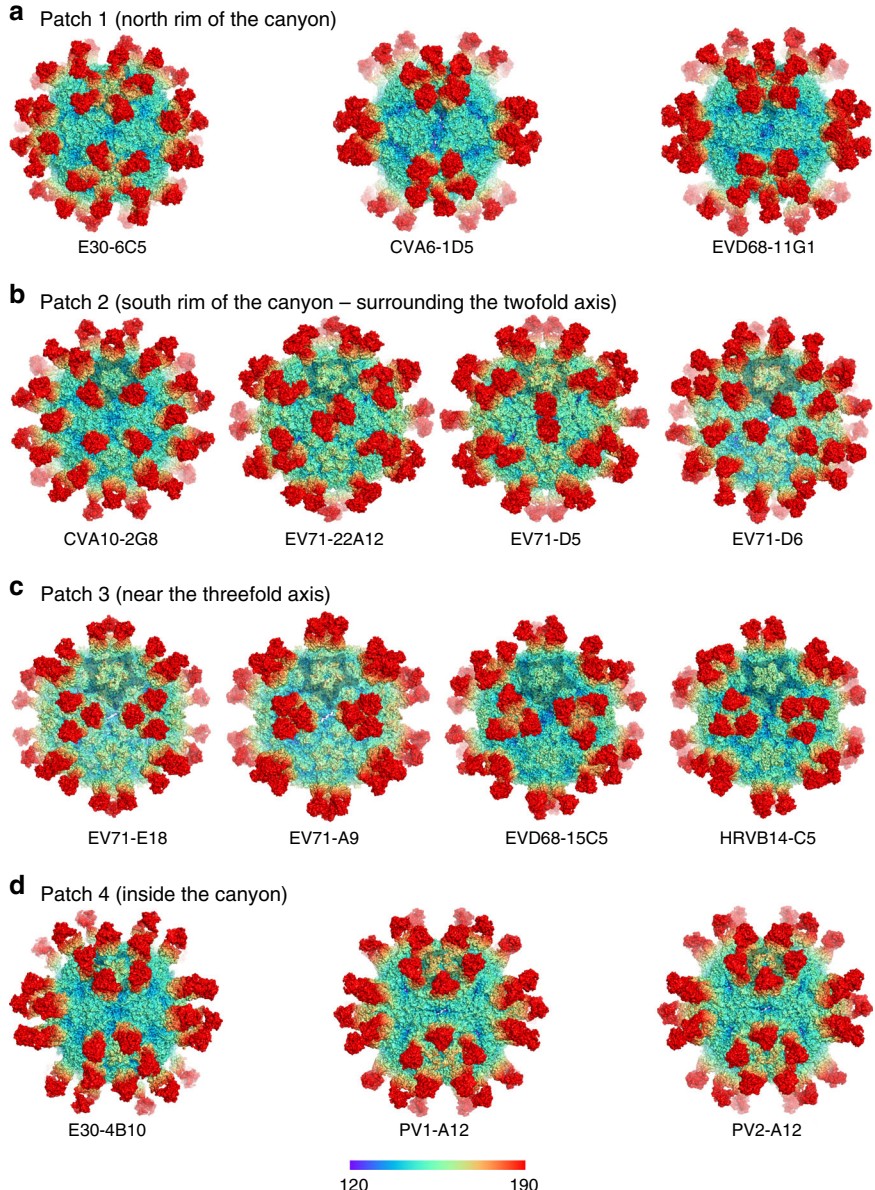

**a** Patch 1 (north rim of the canyon)

E30-6C5          CVA6-1D5          EVD68-11G1

**b** Patch 2 (south rim of the canyon – surrounding the twofold axis)

CVA10-2G8     EV71-22A12     EV71-D5     EV71-D6

**c** Patch 3 (near the threefold axis)

EV71-E18     EV71-A9     EVD68-15C5     HRVB14-C5

**d** Patch 4 (inside the canyon)

E30-4B10          PV1-A12          PV2-A12

120          190

**Fig. 6 Classification of the enterovirus-Fab-complexes.** All published complexes of enterovirus and neutralizing MAb were utilized for comparison. The complexes can be classified into four patches, **a** patch 1—around the north rim of the canyon, including E30-6C5, CVA6-1D5 (PDB CODE: 5XS7) and EVD68-11G1 (PDB CODE: 6AJ9), **b** patch 2—around the south rim of the canyon, which includes EV71-D6 (PDB CODE: 5ZUD), CVA10-2G8 (PDB CODE: 6AD0), EV71-22A12 (PDB CODE: 3J91) and EV71-D5 (PDB CODE: 3JAU), **c** patch 3—near the threefold axis constituted by EV71-E18 (PDB CODE: 4C0U), EV71-A9 (PDB CODE: 5ZUF), EVD68-15C5 (PDB CODE: 6AJ7) and HRVB14-C5 (PDB CODE: 5W3E), and **d** patch 4—inside the canyon, including E30-4B10, PV1-A12 (EMDB CODE: 5670) and PV2-A12 (EMDB CODE: 5671).

onto carbon-coated gold grids freshly glow-discharged. The grids were rinsed, stained with 1% phosphotungstic acid (pH 7.0) and subjected to TEM examination.

**MAbs generation.** The purified formaldehyde-inactivated E30 virions were mixed 1:1 with Freund's complete adjuvant and injected intraperitoneally (ip) into six BALB/c mice at a concentration of 50 μg/mice. All the mice were administered another three doses of booster immunizations with an interval of two weeks between each injection. Five days after the last injection, the spleen cells of the mice were removed and fused with SP2/0 cells to generate hybridomas, followed by cultivation and selection in HAT medium first and HT medium then. Finally, the positive hybridomas were screened by ELISA, where the purified E30 particles were used as coating antigen and each supernatant obtained from the selected hybridomas was used as the primary antibody to screen the ones able to bind with E30 specifically by the methods depicted below. After that, the positive clones were mixed with E30 virions [plaque-forming unit (PFU) ranged from 50 to 100] at 37 °C for 1 h and then added onto the RD cell monolayers to screen the ones that could neutralize E30 by the method of plaque-reduction neutralization test (PRNT) specified below.

**Production of Fab fragments.** MAbs 6C5 and 4B10 were purified using protein A affinity column (GE) from mouse ascites and further used for Fab fragment generation using the Pierce FAB preparation kit (Thermo Scientific) according to the manufacturer's instructions as previously described[11]. Briefly, the samples were first applied to desalination columns to remove the salt and then they were mixed with papain to cleave Fab fragments from the whole MAbs. Finally, the Fab fragments were obtained in the buffer of PBS (ThermoFisher, catalog #10010023) by using the protein A affinity column.

**Cryo-EM and data collection.** For cryo-grids preparation, E30 F-particles were mixed with 6C5 Fab or 4B10 Fab (at a ratio of ~1:200) on ice for 10 s. Afterwards, 3 μl aliquot of the mixture of E30 with 6C5 Fab or a mixture of E30 with 4B10 Fab were deposited onto freshly glow-discharged 400-mesh holey carbon-coated gold grid (Quantifoil, R 1.2/1.3, Jena). Grids were blotted for 3 s in 100% relative humidity for plunge-freeing (Vitrobot; FEI) in liquid ethane. Cryo-EM data sets were collected at 300 kV with a Titan Krios microscope (FEI). Movies (25 frames, each 0.2 s, total dose of 30 e⁻ Å⁻²) were recorded using a K2 detector with a defocus range of 1.2–2.5 μm. Automated single-particle data acquisition was

performed by SerialEM, with a calibrated magnification of 59,000 yielding a final pixel size of 1.32 Å.

**Image processing**. A total of 1,661 and 998 micrographs were recorded for the E30-6C5 and E30-4B10 mixtures, respectively. Out of these, 1,454 and 719 micrographs with visible CTF rings beyond 1/5 Å in their spectra were selected for further processing. The defocus value for each micrograph was determined using Gctf[54]. Then particles were picked and extracted for two-dimensional alignment and three-dimensional reconstructions. The final models of E30-6C5-complex and E30-4B10-complex were generated using 10, 127, and 1,841 particles, respectively. After the high-resolution refinement and postprocessing (estimate the B-factor automatically), the final resolution was evaluated on the basis of the gold-standard Fourier shell correlation (threshold = 0.143)[55]. All the above-mentioned procedures were performed using Relion 3.0[56]. The local resolution was evaluated by ResMap[57]. Statistics related to the data sets and refinement are summarized in Supplementary Table 1.

**Model building and refinement**. The structure of E30 F-particle (see coordinated submission by Wang et al.[15]) was manually fitted into the refined map of E30-Fab-complexes using Chimera[58] and corrected according to the amino sequence of E30 using COOT[59]. The atomic model was built after iterative positional B-factor refinement in real space using PHENIX[60] and re-adjustment in COOT. Likewise, the 6C5/4B10 Fab fragment was corrected from the crystal structure of R10 (PDB ID: 5WTG) and further combined with the viral portion structure to finally obtain the atomic structures of the complexes.

**Enzyme-linked immunosorbent assay (ELISA)**. The purified particles (E30, E6, E3, E11 or CVB3) were coated onto ELISA plates (Costar, Corning, USA) at 30 ng/well, followed by overnight incubation at 4 °C. The coated plates were blocked with 1% BSA in PBST (PBS plus 0.1% Tween 20) at 37 °C for 2 h. Thereafter, the plates were washed five times with PBST and 6C5-IgG or 4B10-IgG was added as primary antibody to each well at 10-fold serial dilutions resulting in a range of concentrations from 0.02 nM to 20 nM. The plates were kept at 37 °C for 1 h after which they were washed again with PBST five times. HRP-conjugated goat anti-mouse IgG H&L (# AP308P, 1/3,000 dilution) (Sigma-Aldrich, St. Louis, USA) was added as a secondary antibody and the plates were incubated at 37 °C for 0.5 h. The plates were washed with PBST five times, and 3,3′,5,5′ -tetramethylbenzidine (TMB) substrate (# P0209, Beyotime, Shanghai, China) was added to each well for 5 min at room temperature. Finally, 2 M $H_2SO_4$ was added to the plates to stop the reaction, and the absorbance value of each well was read at 450 nm.

**Surface plasmon resonance**. Surface plasmon resonance (SPR) experiments were performed using a BIAcore T100 machine (Biacore, GE Healthcare) in PBS buffer (supplemented with 0.05% Tween-20) at 25 °C. The purified E30-F particles were immobilized onto CM5 sensor chip surface using the NHS/EDC method such that a response unit of ~800 was obtained. Then gradient concentrations of 6C5 or 4B10 were passed over the chip at a rate of 20 μl/min. A solution containing 10 mM glycine-HCl (pH 1.7) was used to regenerate the chip after each injection cycle. The binding affinities were obtained by fitting the curves globally using the BIAevaluation software (version 4.1). For the competition assay, the first sample flew over the chip at a rate of 20 μl/min for 120 s, then the second sample was injected at the same rate for another 120 s. The response units were recorded and analyzed using the same software as mentioned above.

**Plaque-reduction neutralization test**. The 6C5-IgG, 4B10-IgG, 6C5-Fab, 4B10-Fab or mice sera were diluted in Dulbecco's Modified Eagle's medium (DMEM) to reach a series of 2-fold dilutions with the highest concentrations of 128 nM, 128 nM, 384 nM, 348 nM or 1:64, respectively. We also set up a control group without any antibody, Fab or sera. The same number of E30 virions (PFU ranged from 50 to 100) were mixed with each dilution, incubated at 37 °C for 1 h and added to the confluent monolayers of RD cells seeded in 6-well plates. The plates were put back in the cell incubator with gentle rock every 20 min for 1 h, followed by three rinses with DMEM (pH 7.4). Then they were covered with the agarose overlay (2 mL/well) supplemented with 2% FBS and further incubated in the 5% CO2 cell incubator at 37 °C for 3 days. Plaques were visualized by staining with 2.5% crystal violet and the percent of inhibition was calculated as $(N_{control} - N_{test})/N_{control} \times 100\%$, where $N_{control}$ and $N_{test}$ represent the mean of plaque counts observed in the control group and test group, respectively. All experiments were performed in triplicate.

**Epitope competition assay**. To test whether the epitopes bound with antibody 6C5 or 4B10 were immunodominant or not, we first coated the purified E30 F-particles onto Elisa plates and blocked the plates as described above in the Elisa method. Sera obtained against aluminum hydroxide-adjuvanted E30 F-particle, E-particle, or aluminum hydroxide were added. A negative control containing the aluminum hydroxide-immunized sera was also set up under similar conditions.

After incubation at 37 °C for 0.5 h, all the wells were washed five times with PBST. HRP-conjugated 6C5/4B10-IgG was added and the plates were incubated for a further period of 0.5 h at 37 °C. Then the wells were washed with PBST and the TMB substrate was added. After 5 min at room temperature, 2 M $H_2SO_4$ was added to stop the reaction. Plates were read at A450 nm. The percent of inhibition was calculated as $(OD_{negative} - OD_{sera})/OD_{negative} \times 100\%$.

**Particle stability thermal release assay**. PaSTRy was performed with SYTO9 or SYPRO Red (Invitrogen, Carlsbad, USA) as fluorescent probes to detect the exposed RNA or hydrophobic residues by an MX3005 qPCR instrument (Agilent, Santa Clara, USA). Here, we set up a pH = 7.4 50 μl-reaction system which contained 2 μg of target protein i.e., E30 alone, or E30 and 6C5, or E30 and 4B10, or E30 pre-heated at 52 °C for 5 min, 5 μM of SYTO9 or 3x SYPRO Red, and ramped up the temperature from 25 °C to 99 °C. Fluorescence was recorded in triplicate at an interval of 1 °C.

**Real-time RT-PCR**. The amount of E30 remaining on the surface of RD cells after 6C5/4B10 treatment was quantitated by real-time RT-PCR. Briefly, E30 was mixed with various concentrations of 6C5 before and after the virus attachment to cells with a MOI of 1 at 4 °C. Then cells were washed three times and their total RNA was extracted by RNeasy mini kit (Qiagen, Hilden, Germany) and subjected to qPCR using SuperScrip III Platinum SYBR Green One-Step qRT-PCR Kit (Invitrogen, Carlsbad, USA) on the QuantStudio Dx Real-Time PCR Instrument (Applied Biosystems, Foster City, USA). The 20 μL reaction mixture contained 0.2 μL SuperScript III RT/Platinum Taq Mix, 5 μL 2X SYBR Green Reaction Mix, 0.2 μL each of 10 μM forward (5′-AAC AGC GTT GCC CGC GTC TA-3′) and reverse primers (5′-ACC CTG TAG TTC CCT ACA TA -3′), 2 μL total RNA, and 2.4 μL RNase-free H2O. The thermal profile for qPCR was as follows—42 °C for 5 min for reverse transcription, 95 °C for 5 min for reverse transcription inactivation; this was followed by 40 cycles of denaturation at 95 °C for 15 s, annealing and extension at 60 °C for 30 s. Endogenous housekeeping gene β-Actin (forward 5′-GCC CTG AGG CAC TCT TCC A-3′, reverse 5′-CGG ATG TCC ACG TCA CAC TT-3′) was used as an internal control to normalize samples. The analysis of relative levels of E30 RNA in different samples was performed by employing the comparative $2^{-\Delta\Delta Ct}$ method[61].

**Reporting summary**. Further information on experimental design is available in the Nature Research Reporting Summary linked to this paper.

## Data availability

The atomic coordinates of E30 (F-particle)-6C5-complex, E30 (F-particle)-4B10-complex and E30 (E-particle)-6C5-complex have been submitted to the Protein Data Bank with accession numbers: 7C81, 7C80 and 7CMK, respectively. The cryo-EM density maps of E30 (F-particle)-6C5-complex and E30 (F-particle)-4B10-complex have been deposited in the Electron Microscopy Data Bank under accession codes: EMD-30304, EMD-30303 and EMD-30408, respectively. Source data are provided with this paper. Other data are available from the corresponding authors upon reasonable request. Source data are provided with this paper.

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

## Acknowledgements

We thank B. Zhu, X. Huang and G. Ji for Cryo-EM data collection at the Center for Biological imaging (CBI), Institute of Biophysics, and X. Pei for sample screening and data collection in the cryo-EM platform of PKU. We also thank Y. Chen, Z. Yang and B. Zhou for SPR technical support and X. Yu for AUC technical guidance. Work was supported by the Strategic Priority Research Program (XDB29010000), Beijing Natural Science Foundation-Haidian Primitive Innovation Joint fund (19L2008), the Key Programs of the Chinese Academy (KJZD-SW-L05), the National Key Research and Development Program (2018YFA0900801 and 2017YFC0840300), National Science Foundation of China (31800145, 31941011, 31900873 and 81520108019) and Center for Biosafety Mega-Science, CAS and Applied Technology Research and Development Project of Heilongjiang Province (GA19B301). Xiangxi Wang was supported by Ten Thousand Talent Program and the NSFS Innovative Research Group (No. 81921005). Ling Zhu was supported by the Youth Innovation Promotion Association at the Chinese Academy of Sciences (2019098).

## Author contributions

K.W., B.Z., Q.Z., and X.S. performed the experiments; K.W. and X.W. solved the structure, L. Zhang, L.C., Z.G., Y.G., and W.Z. provided reagents, X.W., Z.R., F.Z., L. Zhu, and K.W. designed the study, all authors analyzed data, K.W. and X.W. wrote the paper.

## Competing interests

The authors declare no competing interests.
