## [Peer Review File · Nature Communications]

REVIEWER COMMENTS

Reviewer #1 (Remarks to the Author):

The manuscript by Wang and colleagues is a natural continuation of a companion manuscript describing the Echovirus 30 (E30) interaction with its attachment and uncoating receptors: CD55 and RnFc. The study describes the interactions of E30 with two potent neutralizing antibodies that block capsid binding to the two receptors. Both antibodies attach to the capsid in the canyon region, partially overlapping with receptors binding footprints – an area with high variability across different echoviruses.

While the results are less exciting than the those presented in the first manuscript, the study brings to light important details on the molecular mechanism of antibody neutralization of enteroviruses. Such information is essential for rational designing vaccines with large applicability spectrum.

The manuscript is well constructed, the figures are clear and very informative. However, I want to raise several points that will need to be corrected before publication.

Major point:

What it makes this manuscript particularly valuable for picornavirologists is the comprehensive analysis done with all the neutralizing antibodies against enteroviruses. My suggestion would be to develop that part and maybe introduce in the main text a new figure that would illustrate the ideas presented in lines 301-315.

Minor points:

Line 158: Reverse transcription polymerase chain reaction before using the abbreviation RT-PCR would be nice.

Line 293-295: "These previously reported observations together with our current structural analysis suggest antibodies targeting epitopes "inside the canyon" are capable of conferring cross-protection against EVs". I could be wrong, but I find this to be a strong statement considering 4B10 (binding inside the canyon) didn't provide cross-protection against other EV-B strains in the cell-based viral neutralization assays (Line 131-132). Previous reports and the findings from this paper support the hypothesis they have made in the line 323-326 .

Line 335: 1,500xg should be spaced to 1,500 x g (same issue in lines 336 and 337)

Line 336: the gradient range before the continuous sucrose density is preferred. For example, 10-40%

Line 338: Phosphate-buffered saline before PBS abbreviation

Line 341: droped should be corrected to dropped

Line 342: subject should be subjected

Line 344: Mabs should be MAbs. That's what it was used throughout the paper. It must be consistent.

Line 352: How did the authors obtained these complexes? I assume mixing E30-F particles with the Fabs for 30 mins/1 h at 4 °C. I think they should mention it somewhere here.

Line 356: correct close to dose

Line 370: Please indicate the version of the Relion.

Line 411: plaque-forming unit before the PFU abbreviation

Line 433: Please use Particle stability thermal release assay instead of thermal stability assay. This indicates the thermal stability assay we know (using SYTO9, SYPRO Orange/Red). Otherwise there are some other microscale fluorescent thermal stability assays with other dyes.

Line 444: subject should be corrected to subjected

Please add spaces between numbers and units throughout the methods. Especially between the number and °C – multiple cases in the methods section.

Fig1: Please increase the vertical spacing between the figures? They seem too clustered.

Fig2c: bottom left: CD55, 1A1, ICAM5 labels are too close to each other-almost overlapped.

Fig3c and d: please correct "pentomer" to pentamer

Supp fig 1 c: One on the left is SYTO9 and on the right is SYPRO-Red. It should be mentioned in the figure legend. The authors failed to mention about SYPRO-Red in the methods.

Line 844: kD should be kDa and introduce a space between numbers and units.

Reviewer #2 (Remarks to the Author):

This study immunized the mice and elicited two highly potent serotype-specific neutralizing Abs, referred to as 6C5 and 4B10, both of which could neutralize E30 infection efficiently by blocking viral binding to its two types of receptors. The SPR assays showed the competitive binding results, both 6C5 and 4B10 efficiently prevented E30 attachment to the cell surface and could displace the viral particles that had already attached to the cell surface in a dose-dependent manner. Using the Cryo-EM micrographs to see the E30 in complex with its NAbs 6C5 and 4B10 and define the key amino residues for the antigen-antibody interaction.

This study is very systematic and well define the specificity of 6C5 and 4B10 to E30.

The results have showed the main effect of mAbs neutralizing activity depended by blocking receptor binding and not destabilization of the viral structure.

In Figure 3, there are no RNA released after the E30 incubated with mAbs. It still requires a positive/ negative control to show the E30 released the RNA. For example, you could show the E30 released the RNA in 1-3 °C or any condition can make the E30 release the RNA. We would like to see the difference and make sure the assay is well done.

It is very interesting to heard that the Abs destabilized the virions at low temperature. Would it be possible to show the result of intact antibodies and Fab fragments of 6C5 and 4B10 destabilized E30 virions by 1-3 °C. This observation is very different with EV71. In general, the EV71 were uncoated by long incubation time at 37 °C or bound with mAbs for a short incubation time.

Please show your result or explain what the differences in between, thanks.

The supplement figures 1.C were not labeled clearly for E30 complex with mAb, please make the label clearly and describe more in detail because we did not know the real means in the figures.

Manuscript Title: "Serotype specific epitopes identified by neutralizing antibodies underpin immunogenic differences in Enterovirus B"

Response to referees' comments

We thank the reviewers for their positive and constructive comments, and we believe that after incorporating the reviewer's suggestions the manuscript has been strengthened.

Reviewer #1 (Remarks to the Author):

The manuscript by Wang and colleagues is a natural continuation of a companion manuscript describing the Echovirus 30 (E30) interaction with its attachment and uncoating receptors: CD55 and RnFc. The study describes the interactions of E30 with two potent neutralizing antibodies that block capsid binding to the two receptors. Both antibodies attach to the capsid in the canyon region, partially overlapping with receptors binding footprints – an area with high variability across different echoviruses.

We thank the reviewer for a comprehensive evaluation of our two closely related studies on E30, ranging from viral entry to neutralization mechanisms.

While the results are less exciting than the those presented in the first manuscript, the study brings to light important details on the molecular mechanism of antibody neutralization of enteroviruses. Such information is essential for rational designing vaccines with large applicability spectrum. The manuscript is well constructed, the figures are clear and very informative. However, I want to raise several points that will need to be corrected before publication.

We thank the reviewer for a very high evaluation of our first work on molecular basis for E30 entry and considering the two closely related studies as important contributions in the fields of virology and translational medicine.

Major point:

What it makes this manuscript particularly valuable for picornavirologists is the comprehensive analysis done with all the neutralizing antibodies against enteroviruses. My

suggestion would be to develop that part and maybe introduce in the main text a new figure that would illustrate the ideas presented in lines 301-315.

Many thanks for your suggestions. Agreeably, comprehensive comparisons and analysis of the neutralizing antibodies against enteroviruses are of importance for picornavirologists. We have moved the Supplementary Figure 4 (analysis of the neutralizing antibodies) to the main manuscript (now Figure 6) and expanded the analysis a little bit to illustrate these ideas in lines 301-315.

Minor points:

Line 158: Reverse transcription polymerase chain reaction before using the abbreviation RT-PCR would be nice.

Thanks for your suggestion, done.

Line 293-295: "These previously reported observations together with our current structural analysis suggest antibodies targeting epitopes "inside the canyon" are capable of conferring cross-protection against EVs". I could be wrong, but I find this to be a strong statement considering 4B10 (binding inside the canyon) didn't provide cross-protection against other EV-B strains in the cell-based viral neutralization assays (Line 131-132). Previous reports and the findings from this paper support the hypothesis they have made in the line 323-326.

Thanks for pointing this out. Sorry, we did not state this point as clearly as we should have done; this has led to some misunderstanding. We have corrected our statement here as follows "antibodies targeting epitopes "inside the canyon" have the potential to confer cross-protection against EVs and these "in-canyon" antibodies may largely mimic receptor binding mode".

Line 335: 1,500xg should be spaced to 1,500 x g (same issue in lines 336 and 337)

Thanks, corrected!

Line 336: the gradient range before the continuous sucrose density is preferred. For example, 10-40%

Thanks for pointing this out. Details about the range of the density of the sucrose used in the column have been provided in the revised manuscript.

Line 338: Phosphate-buffered saline before PBS abbreviation

Thanks, done.

Line 341: dropped should be corrected to dropped

Thanks, corrected.

Line 342: subject should be subjected

Thanks, corrected.

Line 344: Mabs should be MAbs. That's what it was used throughout the paper. It must be consistent.

Thanks for your suggestions, done!

Line 352: How did the authors obtained these complexes? I assume mixing E30-F particles with the Fabs for 30 mins/1 h at 4 °C. I think they should mention it somewhere here.

Thanks for pointing this out. More details on how to prepare these complexes have been provided in the revised version as follows - "E30 F-particles were mixed with 6C5 Fab or 4B10 Fab (at a ratio of ~1:200) on ice for 10 s. Afterwards, a 3 µl aliquot of the mixture of E30 with 6C5 Fab or a mixture of E30 with 4B10 Fab.....".

Line 356: correct close to dose in line

Thanks, corrected.

Line 370: Please indicate the version of the Relion.

Many thanks for pointing this out. Information about the version of the Relion has been properly provided.

Line 411: plaque-forming unit before the PFU abbreviation

Thanks, done.

Line 433: Please use Particle stability thermal release assay instead of thermal stability assay. This indicates the thermal stability assay we know (using SYTO9, SYPRO Orange/Red). Otherwise there are some other microscale fluorescent thermal stability assays with other dyes.

Thanks for your suggestion, done.

Line 444: subject should be corrected to subjected

Thanks, done.

Please add spaces between numbers and units throughout the methods. Especially between the number and °C – multiple cases in the methods section.

Thanks for pointing this out; done!

Fig1: Please increase the vertical spacing between the figures? They seem too clustered.

Thanks, done.

Fig2c: bottom left: CD55, 1A1, ICAM5 labels are too close to each other-almost overlapped.

Thanks, done.

Fig3c and d: please correct "pentomer" to pentamer

Thanks, corrected.

Supp fig 1 c: One on the left is SYTO9 and on the right is SYPRO-Red. It should be mentioned in the figure legend. The authors failed to mention about SYPRO-Red in the methods.

Thanks for your suggestions. More information has been provided in the figure legends and methods section of the revised manuscript.

Line 844: kD should be kDa and introduce a space between numbers and units.

Thanks, corrected.

Reviewer #2

This study immunized the mice and elicited two highly potent serotype-specific neutralizing Abs, referred to as 6C5 and 4B10, both of which could neutralize E30 infection efficiently by blocking viral binding to its two types of receptors. The SPR assays showed the competitive binding results, both 6C5 and 4B10 efficiently prevented E30 attachment to the cell surface and could displace the viral particles that had already attached to the cell surface in a dose-dependent manner. Using the Cryo-EM micrographs to see the E30 in complex with its NAbs 6C5 and 4B10 and define the key amino residues for the antigen-antibody interaction. This study is very systematic and well define the specificity of 6C5 and 4B10 to E30.

We thank the reviewer for a very comprehensive evaluation of our work.

The results have showed the main effect of mAbs neutralizing activity depended by blocking receptor binding and not destabilization of the viral structure. In Figure 3, there are no RNA released after the E30 incubated with mAbs. It still requires a positive/ negative control to show the E30 released the RNA. For example, you could show the E30 released the RNA in 1-3 °C or any condition can make the E30 release the RNA. We would like to see the difference and make sure the assay is well done.

Many thanks for your suggestions. The particle stability thermal release assay revealed that E30 displays the stability of a typical enterovirus and releases its RNA genome at ~55 °C. Both intact antibodies and Fab fragments of 6C5 and 4B10 destabilized E30 virions a little bit (by 1-3 °C; start to release RNA genome at 52-54 °C), suggesting that E30 continues to exist as a mature virion at physiological temperature even after destabilization by 6C5/4B10. Therefore destabilization is unlikely to be the neutralization mechanism. In addition, the structures of E30 in complex (prepared at low temperature) with 6C5/4B10 showed bulk densities for RNA genome inside (supplementary figure 3), which are similar to those observed in E30 mature virion structure, reflecting that binding of 6C5/4B10 does not trigger viral RNA release for E30. We completely agree that a positive/negative control is required to verify the assay well set or not. As suggested, we demonstrate that heat-treatment at 55 °C for 10 min makes E30 release RNA genome (supplementary figure 2), which can be detected by the dye SYTO9 at room temperature. The positive (heat-treatment) and negative (E30 alone) controls as well as experimental groups have been re-analyzed and this information is provided in supplementary figure 2.

It is very interesting to hear that the Abs destabilized the virions at low temperature. Would it be possible to show the result of intact antibodies and Fab fragments of 6C5 and 4B10 destabilized E30 virions by 1-3 °C. This observation is very different with EV71. In general, the EV71 were uncoated by long incubation time at 37 °C or bound with mAbs for a short incubation time. Please show your result or explain what the differences are between, thanks.

Sorry, we may have not explained this point clearly, which has led to some misunderstanding. Related with the previous point, both antibodies - 6C5 and 4B10 - slightly destabilized E30 virions by 1-3 °C (not at 1-3 °C). The particle stability thermal release assay and complex structure (prepared on ices) analysis reveals these two antibodies DO NOT destabilize E30 virions at low temperature. We have modified our statement on this point to avoid possible misunderstandings as follows “However, both intact antibodies and Fab fragments of 6C5 and 4B10 destabilized E30 virions slightly by 1-3 °C (Fig. 1h and Supplementary Fig. 1). Given that E30 virions continue to exist as mature virion at physiological temperatures even after slight destabilization by 6C5 and 4B10 (Supplementary Fig. 2), destabilization of the virus is unlikely to be their mechanism of neutralization”. Based on previous studies on EV71 and HRV, a number of neutralizing antibodies capable of initiating genome release (as

mentioned by the reviewer), in general, destabilize the virion substantially and dissociate viral capsid protein interactions to trigger viral uncoating. Analysis of different neutralization mechanisms played by various neutralizing antibodies has been discussed in the revised manuscript.

The supplement figures 1.C were not labeled clearly for E30 complex with mAb, please make the label clearly and describe more in detail because we did not know the real means in the figures.

Many thanks for your suggestions. We have modified the labels and provided more details in figure legends to present the information clearly.

REVIEWERS' COMMENTS:

Reviewer #2 (Remarks to the Author):

This study immunized the mice and elicited two highly potent serotype-specific neutralizing Abs, referred to as 6C5 and 4B10, both of which could neutralize E30 infection efficiently by blocking viral binding to its two types of receptors. The SPR assays showed the competitive binding results, both 6C5 and 4B10 efficiently prevented E30 attachment to the cell surface and could displace the viral particles that had already attached to the cell surface in a dose-dependent manner. Using the Cryo-EM micrographs to see the E30 in complex with its NAbs 6C5 and 4B10 and define the key amino residues for the antigen-antibody interaction.

This study is very systematic and well define the specificity of 6C5 and 4B10 to E30.

1. Through the author's explanation, it is clearly to understand that destabilization is unlikely to be the neutralization mechanism. In addition, the structures of E30 in complex (prepared at low temperature) with 6C5/4B10 showed bulk densities for RNA genome inside (supplementary figure 3), which are similar to those observed in E30 mature virion structure, reflecting that binding of 6C5/4B10 does not trigger viral RNA release for E30.

2. The authors have followed the suggestion to set a positive/negative control is required to verify the assay well set or not and demonstrate that heat-treatment at 55 ° C for 10 min makes E30 release RNA genome (supplementary figure 2), which can be detected by the dye SYTO9 at room temperature. The positive (heat-treatment) and negative (E30 alone) controls as well as experimental groups have been re-analyzed and this information is provided in supplementary figure 2.

3. The authors have modified the labels and provided more details in figure legends to present the information clearly supplement figures 1.

4. Accept with no additional suggestions.

I would like to encourage the authors continue to find the answer why the enterovirus antigenicity is so diverse? where is the highly conserved sequences or epitopes among enterovirus? For future universal enterovirus vaccine design.